# Land Use and Ecosystem Services Evolution in Danjiangkou Reservoir Area, China: Implications for Sustainable Management of National Projects

Linghua Liu [1,2], Liang Zheng [3,4], Ying Wang [1,2,*], Chongchong Liu [1,2], Bowen Zhang [1,2] and Yuzhe Bi [1,2]

1   School of Public Administration, China University of Geosciences, Wuhan 430074, China
2   Key Laboratory of Rule of Law Research, Ministry of Natural Resources, Wuhan 430074, China
3   Changjiang Institute of Survey, Planning, Design and Research, Wuhan 430014, China
4   Key Laboratory of Changjiang Regulation and Protection of Ministry of Water Resources, Wuhan 430014, China
*   Correspondence: yingwang0610@cug.edu.cn

**Abstract:** The South-to-North Water Diversion Project (SNWDP) is one of the largest cross-basin and cross-region water transfer projects in the world. The Danjiangkou reservoir area, a haven of diverse species, serves as a core water source for the Central Line of the SNWDP. Yet, less research has been conducted on changes in land use and ecosystem services (ESs) in the Danjiangkou reservoir area in the context of the implementation of the SNWDP and other national projects. In this study, we aim to reveal evolutions of land uses and ESs in the Danjiangkou reservoir area and the response of ESs to natural and socio-economic factors. This is essential to enhance the regional sustainable management of the Danjiangkou reservoir area. Based on classified land use maps and the InVEST model, we first analyzed the land use changes and evaluated three typical types of ESs (i.e., water yield (WY), carbon storage (CS) and habitat quality (HQ)) in the Danjiangkou reservoir area during 2000 to 2018. Then, we detected the spatial clustering characteristics and tradeoffs and synergistic relationships of multiple ESs through hot spot analysis and correlation analysis. Finally, we adopt the geographical detector model (GDM) to identify key driving factors of ESs changes. The results show that: (1) During 2000–2018, the area of arable land and woodland decreased by 1.65% and 0.8%, respectively, while the proportion of construction land and water area increased by 1.31% and 1.39%, respectively. (2) The greatest decrease was in WY, decreasing by 59%, while the change in HQ was relatively stable, but showed spatial heterogeneity. (3) The northern, southern and western districts of the reservoir area showed mainly synergies among multiple ESs, while other regions showed mainly trade-offs. (4) Road network density, proportion of construction land and normalized difference vegetation index are the leading factors for ESs variations. These findings can provide reference for formulating more reasonable ecological protection strategies, so as to realize the sustainable management of SNWDP and its headwaters region.

**Keywords:** ecosystem services (ESs); south-to-north water diversion project (SNWDP); InVEST model; tradeoffs and synergies; geographical detector model (GDM); Danjiangkou reservoir area; China

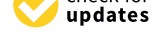



## 1. Introduction

With socio-economic development and continuous population growth on a global scale, the structure and configuration of land use have changed dramatically [1], giving rise to resources shortage [2] and ecosystems degradation [3–5]. Since the introduction of the Millennium Ecosystem Assessment [6], it has further raised the importance of ecological security for domestic and foreign related institutions and strengthened the exploration and research of ecosystem services (ESs) by domestic and foreign experts. This has made the emergence of ESs a hot spot for interdisciplinary research. It is essential to understand the changing trends and driving factors of ESs, especially for ecologically fragile areas or

important ecological functional areas. The reason is that ESs can link ecosystem structure, process, function and human benefits, and provide a framework for understanding how natural and socio-economic systems interact. Thus, further understanding the linkage between land use/land cover (LULC) and ESs is essential for the sustainable management of crucial ESs, especially for ecologically fragile or important regions that may affect ecological security on a global scale.

Influenced by the complex climate and topography, the regional distribution of China's water resources is highly heterogeneous, generally high in the south and low in the north [7]. To redistribute water resources spatially, China implemented the worlds' largest cross-basin water transfer project, i.e., the SNWDP [8,9], aiming to relieve water pressure in North and Northwest China by transferring water from the Yangtze River. Danjiangkou reservoir area, as the water source of for the Central Line of SNWDP, has experienced frequent soil and water alternation and periodic artificial water storage during the construction of the project. These processes have altered regional land use/land cover (LULC) and exerted profound impacts on regional ESs provisions [10,11]. Due to its significant ecological value, another nationwide program for enhancing ESs, i.e., Sloping Land Conversion Program (SLCP) has been implemented in the Danjiangkou reservoir area since 2001. With the aim of increasing vegetation cover [12], the SLCP provides payment incentives to farm households who convert marginal sloping crops to woodlands or grasslands. Ecosystem changes in Danjiangkou reservoir can affect the surrounding areas and even the entire project. Therefore, it would be beneficial to explore how LULC and ESs change in the Danjiangkou reservoir area under the confounding impacts of SNWDP and SLCP, as well as to identify the factors driving ESs changes, as this will assist with subsequent SNWDP Central Line planning and management.

Ecosystem services (ESs) are the benefits that humans derive directly or indirectly from ecosystems that support their survival [13,14]. Currently, ecosystem services research has yielded many results, including temporal-spatial changes in LULC and ESs as well as their relationships at different spatial scales, such as global, national, provincial and municipal, urban clusters, river basins, plateaus, and basins [15–22]. However, there are still relatively few articles on ecosystem services in water sources of important water diversion projects. In this study, we chose the InVEST model to evaluate the ESs in the Danjiangkou reservoir area as it is one of the most widely used tools for assessing ESs, has low data requirements [23,24], and is available globally [25]. In addition, the InVEST model breaks through the geographical restrictiveness of traditional assessment methods and effectively reflects the spatial heterogeneity of ESs [26]. Over the past few decades, ES stability has been challenged by natural processes and human disturbances, during which ESs are influenced by multiple factors and exhibit varying degrees of temporal-spatial heterogeneity. Identifying key drivers of ESs change is the basis for sustainable management of ESs. Numerous studies have shown that climate, vegetation type, biodiversity, etc., are important drivers of ESs dynamics [27,28]. Aside from natural factors, ESs are also influenced by human activities and socio-economic factors [29], of which LULC change plays a leading role [30].

According to existing ES assessment studies [31,32], ESs vary by region, are interrelated, and have complex relationships with each other, e.g., tradeoffs and synergies. ES tradeoffs are usually manifested in a reciprocal change in which one ES rises and another declines, while synergies arise when two ESs increase or decrease simultaneously [33–35]. The exploration of tradeoffs and synergistic mechanism among ESs has now become one of the central issues in ESs research [36], including its theoretical basis, temporal-spatial characteristics, driving factors, etc. Spatial mapping analysis [37–39], statistical analysis [40–42], and model simulation [43,44] are mainly used to analyze ESs tradeoffs and synergies. For example, Gonzalez-Redin et al. [39] developed a raster network-based analysis tool called Bayes-GIS for mapping ES tradeoffs and found that tradeoffs are prevalent in areas with high levels of stakeholder engagement. Based on Spearman's correlation analysis, Tian et al. [42] found that the net primary productivity (NPP) traded off with WY and sediment yield (SY),

respectively, while the relationship between WY and SY was synergistic. Since the goal of ecosystem management is to maximize the benefits of multiple ESs for regional sustainable development [45], it is imperative to uncover the spatio-temporal characteristics of tradeoffs and synergistic relationships between ESs, which could provide guidance for stakeholders to optimize ES management [46] and enhance the implementation of the policy [47]. In this study, we aim to identify and analyze the synergies and tradeoffs among multiple types of ESs in the Danjiangkou reservoir area through correlation analysis, as it can recognize the relationship between the temporal dynamics among ESs and is widely adopted in related research [32,48–50].

As the core water source and an ecologically fragile area of the Central Line of SNWDP in China, Danjiangkou reservoir is under severe environmental pressure from rapid economic development and population growth. Therefore, this study analyzes the spatio-temporal changes of land use and ESs and their spatial clustering characteristics in the national key reservoirs in 2000, 2008 and 2018. Due to the complexity of ESs interactions, this study further analyzed the interactions among multiple ESs in the Danjiangkou reservoir to provide support for achieving sustainable development of ESs. In addition, it is crucial to reveal the influence of natural and socio-economic factors on ESs, which is extremely important for achieving sustainable development in the Danjiangkou reservoir area. Therefore, the main aims of this study were to: (1) explore the spatio-temporal evolutionary characteristics of LULC and multiple ESs in the Danjiangkou reservoir area; (2) reveal the spatial clustering characteristics of ESs and the tradeoffs and synergistic relationships among multiple ESs; and (3) identify the main driving factors affecting ESs variations in the Danjiangkou reservoir area. The findings of this study will contribute to a better understanding of the landscape and local ecosystems of the reservoir region as a result of the Central Line of SNWDP. This could provide guidance for the development of reasonable ecological protection strategies, thereby ensuring that the Danjiangkou reservoir area can both ensure a sustainable development of the ecological environment and continue to play a central role in the Central Line of SNWDP.

## 2. Materials and Methods

### 2.1. Overview of the Study Area

The Danjiangkou reservoir area ($32°14'$~$33°48'$ N, $109°25'$~$111°52'$ E) is situated at the intersection of Henan, Hubei and Shaanxi Provinces, straddling Henan and Hubei Provinces, with a total area of $1.79 \times 10^4$ km$^2$ (Figure 1). As Asia's first artificial freshwater lake, Danjiangkou reservoir provides water for the Central Line of SNWDP. It consists of seven counties, including Xichuan and Xixia in Nanyang City of Henan Province, and Danjiangkou, Yunxi, Yunyang, Zhangwan and Maojian in Shiyan City of Hubei Province. The Danjiangkou Reservoir Dam raising project began in 2005 and was raised from 162 m to 176.6 m, bringing the total reservoir capacity to 29.05 billion cubic meters and the inundated area upstream to 144 km$^2$. Additionally, by the end of 2014, the Danjiangkou Reservoir officially began supplying water to Beijing and Tianjin, as well as many cities in Henan and Hebei. In addition, SNWDP construction has generated 435,000 immigrants, including 345,000 migrants in the Danjiangkou reservoir area, and the number of immigrants relocated in that region is the highest in the history of water conservancy immigration in the world.

The reservoir area is located in the upper reaches of the Han River and in the middle and the mid-western part of the Nanyang Basin, with complex geomorphological features. It has an overall topography of low in the southeast and high in the northwest, and is situated in the transition zone between the second and third steps in China, at an altitude of 74–2149 m. Due to the dramatic topographic changes, the Danjiangkou reservoir area has a subtropical monsoon climate with long winters and short summers, rain and heat in the same season, and is the transition zone of the north–south climate divide in China. As a result of the construction of the Danjiangkou Reservoir and the blocking of winter winds by the Qinling Mountains, significant regional climate characteristics have emerged. Precipitation distribution is uneven across the region, decreasing from south to north. The

mean annual temperature of the reservoir area is 14.4–15.7 °C, and the average annual rainfall is 800–1000 mm, mainly in July–September. The Danjiangkou reservoir area has rich vegetation types and high forest coverage, mainly including evergreen broad-leaf forests, coniferous forests and mixed coniferous broad leaved forests, etc., with outstanding water-holding capacity. It is also home to a wide variety of rare plants and animals, making it an important ecological treasure trove in China.

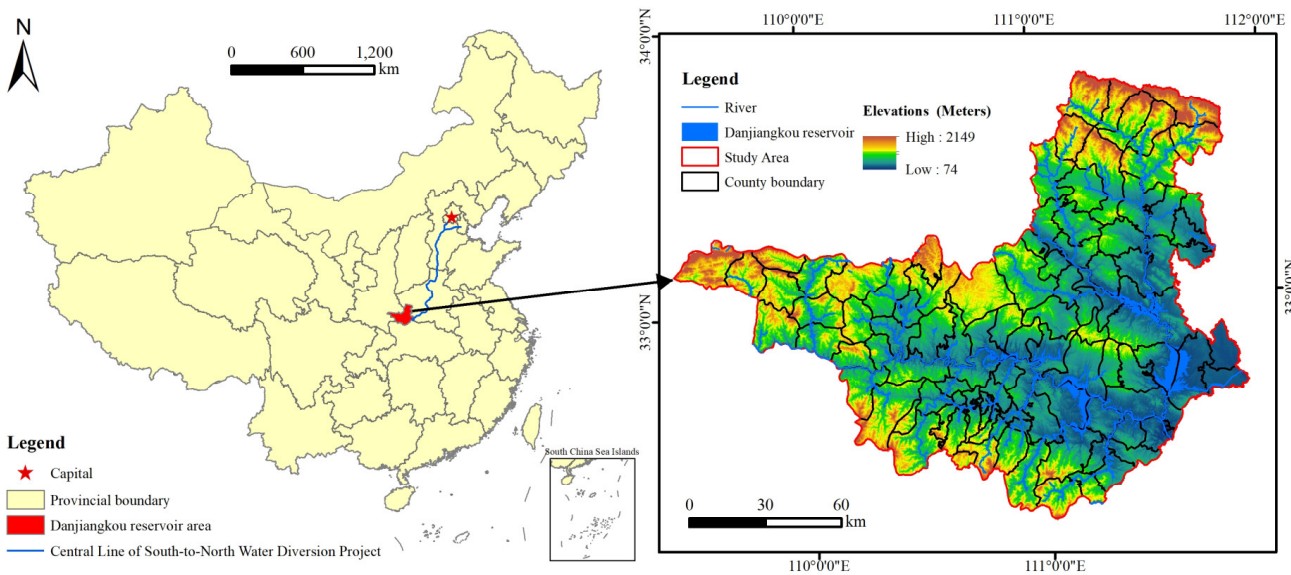

**Figure 1.** Geographic location of the Danjiangkou reservoir area and the Central Line of South-to-North Water Diversion Project.

*2.2. Data Sources*

In the study, the raster data used mainly included land use/land cover (LULC), digital elevation model (DEM), annual precipitation, normalized difference vegetation index (NDVI), China soil database, gross domestic product (GDP), etc. Vector data mainly included rivers, roads and administrative boundaries, etc. LULC datasets was sourced from the Resources and Environmental Sciences and Data Center, Chinese Academy of Sciences, which classified LULC into 29 types with a precision of 30 m × 30 m. We reclassified them into six land use types with a 100 m × 100 m precision, including arable land, construction land, woodland, grassland, water area and unused land. This study used soil data from the Coordinated World Soil Database (HWSD) Chinese soil dataset (v1.1), which mainly includes soil sand content, clay content, silt content, and organic carbon content. We used the analysis tools in ArcGIS 10.5 to calculate the road network density, the river network density, and the distance to the county. In addition, based on the DEM, we computed the slope of Danjiangkou reservoir area through the spatial analysis tool in ArcGIS 10.5 software. Finally, all the raster data needed for the study were converted to the precision of 100 m × 100 m. Specific data types and sources are detailed in Table 1.

**Table 1.** Data sources for this study.

| Type | Data | Source | Precision |
|---|---|---|---|
| Raster | LULC<br>DEM<br>Annual precipitation<br>NDVI<br>GDP | Resources and Environmental Sciences and Data Center, Chinese Academy of Sciences (https://www.resdc.cn/ (accessed on 20 October 2022)) | 30 m<br>30 m<br>1 km<br>1 km<br>1 km |
| | China soil Database | National Tibetan Plateau Data Center (http://data.tpdc.ac.cn (accessed on 22 October 2022)) | 1 km |
| | Potential evapotranspiration | Global Aridity Index and Potential Evapotranspiration (ET0) Climate Database v2 (https://doi.org/10.6084/m9.figshare.7504448.v3 (accessed on 20 October 2022)) | 1 km |
| Vector | Roads<br>Rivers<br>Administrative boundary<br>City location | Resources and Environmental Sciences and Data Center, Chinese Academy of Sciences (https://www.resdc.cn/ (accessed on 20 October 2022)) | /<br>/<br>/<br>/ |

Note: land use/land cover (LULC), digital elevation model (DEM), normalized difference vegetation index (NDVI), gross domestic product (GDP).

*2.3. Methods*

As shown in Figure 2, this study first analyzed the LULC changes in the Danjiangkou reservoir area from 2000 to 2018 and adopted the InVEST model to explore the spatio-temporal changes of three typical ESs in the region, including WY, CS and HQ. Then, the spatial clustering characteristics of ESs were explored through hot spot analysis, and the tradeoffs and synergistic relationships of ESs were further analyzed using correlation analysis. Finally, we adopted GDM to reveal driving factors of ESs changes, including natural and socio-economic factors.

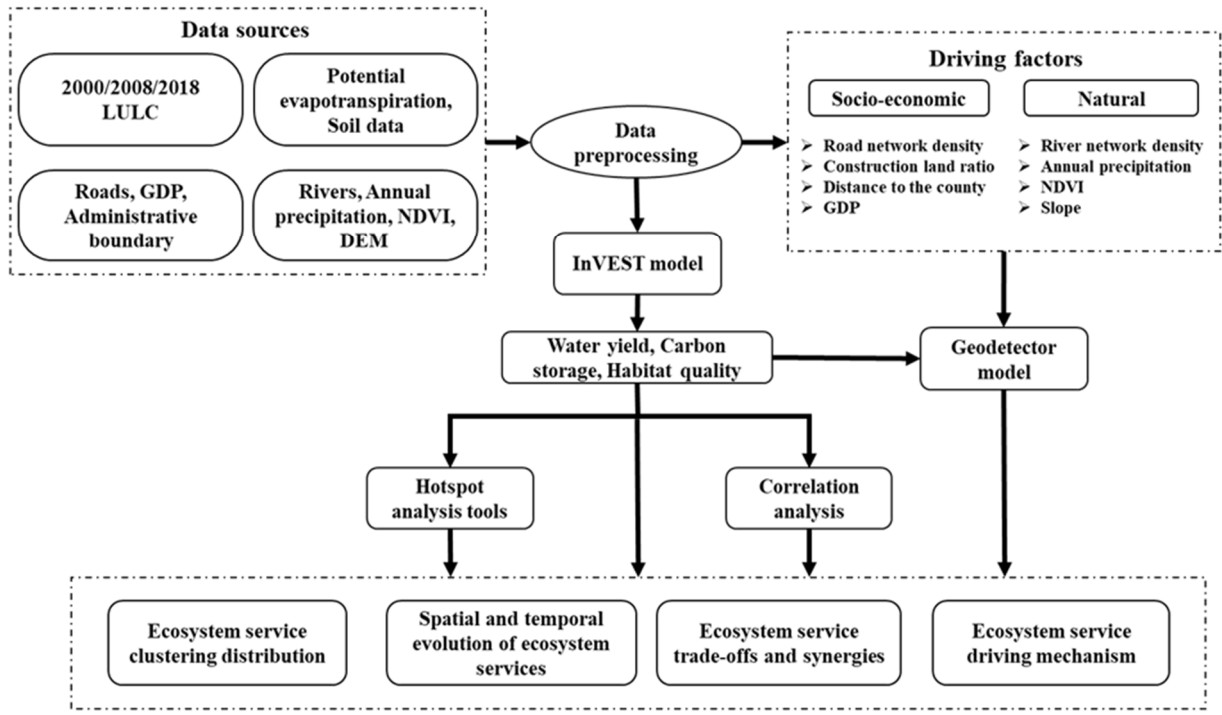

**Figure 2.** Flowchart of this study.

2.3.1. Ecosystem Services Assessment

Maintaining the ecological security of the Danjiangkou reservoir area is of strategic importance since it is the core water source for SNWDP. Therefore, we chose the InVEST

model to assess important ESs in the Danjiangkou reservoir area. According to the Millennium Ecosystem Assessment framework [6], ESs can be categorized into four major types, i.e., provisioning, regulating, supporting, and cultural services. Based on data availability and the most important types of ESs provided by ecosystems in the Danjiangkou reservoir area, we identified three types of ESs, i.e., WY, CS and HQ, which represent provisioning, regulating, and supporting services, respectively. In this study, cultural services are not accounted for due to a lack of data to assess these non-material services that humans obtain from ecosystems.

1. Water Yield (WY)

Based on the principles of water balance, the WY Module in InVEST calculates the WY of each raster cell using annual precipitation, vegetation type, potential evapotranspiration, plant available water content and soil depth as parameters [26]. In this study, it is hypothesized that the WY from each 100 m × 100 m raster is pooled as runoff to the watershed outlet without subdivision of surface and subsurface runoff. Following is the calculation formula:

$$Y_{xj} = \left(1 - \frac{AET_{xj}}{P_x}\right) \times P_x \tag{1}$$

where $Y_{xj}$ and $AET_{xj}$ are the annual WY depth and annual evapotranspiration of element $j$ on a raster $x$, respectively. $P_x$ is the amount of annual precipitation on a raster $x$.

$$\frac{AET_{xj}}{P_x} = \frac{1 + \omega_x + R_{xj}}{1 + \omega_x R_{xj} + \left(\frac{1}{R_{xj}}\right)} \tag{2}$$

$$\omega_x = Z \times \frac{PAWC_x}{P_x} \tag{3}$$

$$R_{xj} = \frac{k_{ij} \times ET_0}{P_x} \tag{4}$$

where $R_{xj}$ is the Budyko drying index of raster $x$ on element $j$. $\omega_x$ is the proportion of the corrected vegetation available water to the expected precipitation. $Z$ is the Zhang coefficient, derived from the seasonal distribution of precipitation. $PAWC_x$ is the available water content of vegetation. $k_{ij}$ is the vegetation evapotranspiration coefficient. $ET_0$ is the potential evapotranspiration.

2. Carbon Storage (CS)

In this study, LULC datasets were used to determine carbon density for each land use type, including aboveground, belowground, soil, and dead biomass. Based on the following formula, the CS module of the InVEST model calculates current CS in the landscape:

$$C_{tot} = C_{above} + C_{below} + C_{soil} + C_{dead} \tag{5}$$

where $C_{tot}$ is the total CS; $C_{above}$, $C_{below}$, $C_{soil}$, and $C_{dead}$ represents CS in aboveground, belowground, soil, and dead biomass, respectively. The specific parameters of the carbon pool table are referred to in the relevant literature [51] and the InVEST user guide [26].

3. Habitat Quality (HQ)

The HQ module was adopted to assess the biodiversity of the study area and to simulate human activities' influence on the habitat. Higher human activity is associated with lower HQ and biodiversity, and vice versa. Different land use types were used to assess the habitat quality of the study area in terms of their threat intensity and sensitivity to threat factors. The spatial weights and impact distances of the threat factors were also considered. The formula is as follows:

$$P_{xj} = H_j \left[1 - \left(\frac{D_{xj}^z}{D_{xj}^z + k^z}\right)\right] \tag{6}$$

where $P_{xj}$ is the HQ index of raster $x$ on element $j$. $H_j$ represents the habitat suitability of element $j$ and has a value of (0, 1). $D_{xj}$ is the degree of habitat degradation of raster cell $x$ in element $j$. $k$ is a constant of half-saturation and has a value of 0.5. $z$ is a default normalization constant of the model, which is usually 2.5.

### 2.3.2. Hot Spot Analysis

Based on the ESs assessment, we further identified the ESs hot spots in the Danjiangkou reservoir area, which was helpful to understand ES supply capacity in different areas within the reservoir area. Cold spots and hot spots are areas with high or low aggregation of ES characteristic values. Hot spot areas require elements that have high values and are surrounded by other elements that also have high values. In spatial statistics, the *Gi\** coefficient proposed by Getis and Ord [52], a local spatial autocorrelation indicator based on the full matrix of distances, is commonly used to identify areas where high or low values of ESs are clustered and distributed in space. In this study, the values of three ESs for the years (2000, 2008, and 2018) were normalized and overlaid for analysis, resampled at the township scale, and the z-scores and *p*-values between each element were calculated to identify the locations where clustering of hot spot or cold spot elements occurred in space. The formula for calculating the $Gi^*$ coefficient is as follows:

$$G_i^* = \frac{\sum_j^n W_{ij}x_j}{\sum_j^n x_j} \tag{7}$$

normalize $G_i^*$ to obtain the $Z$ value:

$$Z(G_i^*) = \frac{G_i^* - E(G_i^*)}{\sqrt{VAR(G_i^*)}} = \frac{\sum_j W_{ij}x_j - \bar{x}\sum_j^n W_{ij}}{s\sqrt{\frac{n\sum_j^n W_{ij}^2 - \left(\sum_j^n W_{ij}\right)^2}{n-1}}} \tag{8}$$

$$\bar{X} = \frac{\sum_j^n x_j}{n} \tag{9}$$

$$S = \sqrt{\frac{\sum_j^n x_j^2}{n} - \left(\bar{X}\right)^2} \tag{10}$$

where $G_i^*$ is the z-score, $w_{ij}$ is the spatial weight matrix between element $i$ and element $j$. $\bar{X}$ is the average of all elements. $x_j$ is the attribute value of element $j$ and $n$ is the total number of elements.

### 2.3.3. Tradeoff and Synergy Analysis

To examine the synergies and trade-offs among the three typical types of ESs from 2000 to 2018, this study used Pearson correlation coefficients and spatial analysis tools in ArcGIS 10.5 for spatial identification and analyses. The formula is as follows:

$$\rho_{XY} = \frac{Cov(X,Y)}{\sqrt{D(X)}\sqrt{D(Y)}} \tag{11}$$

where $\rho_{XY}$ is the correlation coefficient between $X$ and $Y$ and has a value of $(-1, 1)$. When both $X$ and $Y$ increase or decrease, there is a positive correlation between $X$ and $Y$. *Pearson* correlation coefficients within the range of (0, 1) indicate synergy. When $X$ increases (decreases) and $Y$ decreases (increases), there is a negative correlation between $X$ and $Y$. A value of *Pearson* correlation coefficient in the range of (–1, 0) indicates the existence of a tradeoff relationship. In particular, $0 < \rho_{XY} \leq 0.5$ was classified as low synergy and $0.5 < \rho_{XY} \leq 1$ as high synergy; while $-0.5 \leq \rho_{XY} < 0$ was classified as low trade-off and $-1 \leq \rho_{XY} < -0.5$ as high trade-off.

2.3.4. Geographical Detector Model (GDM)

This paper uses a GDM to detect the effects of multiple factors on ESs. The GDM can detect spatial variations and identify potential influential factors, which can reveal similarities and differences between regions [53]. The GDM mainly includes interaction detection, factor detection, ecological detection and risk detection. This study focuses on the factor detection module of the GDM. The factor detector module is applied to examine the degree to which an explanatory variable factor explains the outcome variable, and it is usually identified and analyzed using q-values. The following is the calculation formula:

$$q = 1 - \frac{1}{N\sigma^2}\sum_{h=1}^{L} N_h\sigma_h^2 \tag{12}$$

where q denotes the explanatory power of the explanatory variable on the outcome variable and takes values in the range (0, 1). A larger q value means a stronger explanation, and vice versa. $h = 1,2,3, ..., L$ represents the classification or partition of the variable.

In this study, we used the factor detector module of GDM to detect and analyze the driving factors of ES variations. The Danjiangkou reservoir area, one of China's most significant ecological areas, is extremely fragile, especially under the impact of human activities, which can lead to irreversible damage. By referring to the relevant literature on the driving mechanisms of ESs [54–56] and considering the actual situation of the Danjiangkou reservoir area [57] and the availability and accessibility of data, eight drivers of ESs were selected from both natural environmental and socio-economic aspects. Socio-economic factors bring the most direct impact on changes in ESs. The study found that land use change has a substantial impact on ESs [58]. In particular, rapid socio-economic development has led to rapid expansion of land for construction, which has caused great damage to the ecological environment. In addition, the proximity of socio-economic centers also influences ESs significantly [56]. Therefore, we selected road network density ($X_1$), construction land ratio ($X_2$), distance to the county ($X_3$), and GDP ($X_4$) among the socio-economic factors. Natural environmental conditions, such as vegetation cover, altitude, and water availability, to determine the ecosystems' ability to provide critical ES [59]. From the perspective of long-term ES management in the Danjiangkou reservoir area, we selected the river network density ($X_5$), annual precipitation ($X_6$), NDVI ($X_7$), and slope ($X_8$) among the natural environmental factors.

## 3. Results

### 3.1. LULC Change

In the Danjiangkou reservoir area, about 60% of the landscape was covered by woodland, followed by arable land and grassland. During 2000–2018, the LULC of the Danjiangkou reservoir area has altered considerably under the influence of human activities. For example, from 2000 to 2018, the area proportion of arable land and woodland decreased by 1.65% and 0.8%, respectively, while the proportion of construction land and water area increased by 1.31% and 1.39%, respectively (Figure 3). Compared with 2000, the spatial distribution of LULC in the Danjiangkou reservoir area in 2018 has changed significantly. There was a noteworthy expansion of water area in eastern areas near the reservoir. Water areas have been expanded most significantly in Xichuan County. This was mainly due to the significant changes brought about by the implementation of the Central Line of SNWDP and the construction of the Danjiangkou Dam. In addition, the expansion of construction land in 2018 compared to 2000 was concentrated in the southern part of the study area in the city of Shiyan. Despite the expansion trend, water area and construction land in the Danjiangkou reservoir are relatively small proportions of its total land use area.

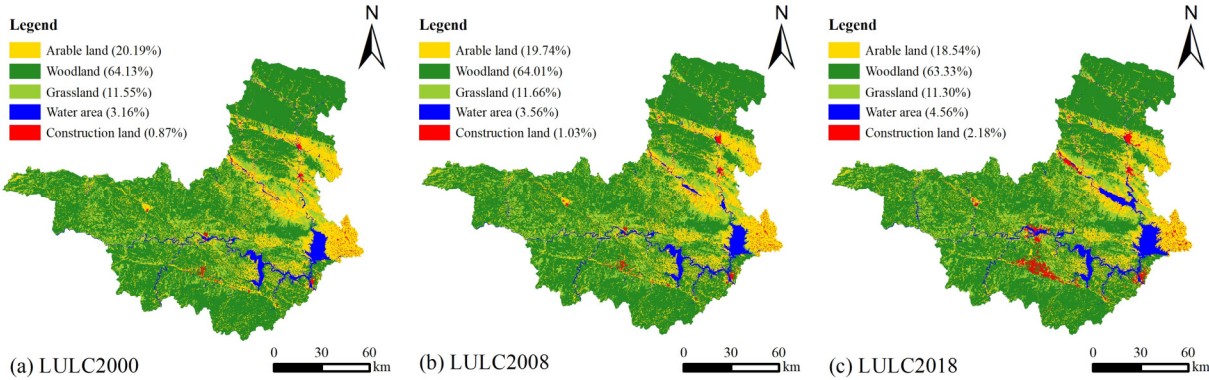

**Figure 3.** Land use type and proportion in the Danjiangkou reservoir area.

According to the land use transition in the Danjiangkou reservoir area from 2000 to 2018 (Table 2), the total area of land that underwent conversion during the study period was 1859.03 km$^2$, accounting for 10.38% of its total area. The change in transition shows that the area of arable land, grassland and woodland has decreased, while the area of water areas and construction land has increased. The total area transferred from arable land and woodland to other types accounted for 80.19% of the total transferred area. The results show that a large part of arable land has been converted to woodland, mainly distributed in mountainous areas, which is largely due to the implementation of SLCP. Over the same period, a large portion of woodland was converted to arable land due to socio-economic development. These areas of inter-conversion of arable land and woodland were mainly located in the western and middle parts of the study area and along the Danjiangkou Reservoir. Another part of the arable land was converted into the water area, which was due to the construction of the Danjiangkou Dam, which greatly expanded the water area of the reservoir, resulting in a large area of arable land along the reservoir being submerged by water. In addition, along with the socio-economic development and urbanization expansion, a large amount of arable land is converted into land for construction.

**Table 2.** Land use transition in the Danjiangkou reservoir area from 2000 to 2018 (km$^2$).

| Year | | **2018** | | | | | | | Decreased |
|------|------|------|------|------|------|------|------|------|------|
| | **LULC** | **Arable Land** | **Woodland** | **Grass Land** | **Water Area** | **Construction Land** | **Unused Land** | **Total** | |
| **2000** | Arable land | 2805.42 | 348.92 | 78.58 | 229.42 | 155.43 | 0.00 | 3617.77 | 812.35 |
| | Woodland | 377.03 | 10,811.58 | 149.63 | 51.95 | 99.57 | 0.24 | 11,490.00 | 678.42 |
| | Grassland | 84.80 | 162.13 | 1791.58 | 14.62 | 17.11 | 0.03 | 2070.27 | 278.69 |
| | Water area | 21.12 | 20.35 | 3.86 | 534.68 | 3.43 | 0.00 | 583.44 | 48.76 |
| | Construction Land | 34.48 | 4.02 | 0.59 | 1.72 | 115.62 | 0.00 | 156.43 | 40.81 |
| | Total | 3322.85 | 11,347.00 | 2024.24 | 832.39 | 391.16 | 0.27 | 17,917.91 | - |
| | Increased | 517.43 | 535.42 | 232.66 | 297.71 | 275.54 | 0.27 | - | 1859.03 |

*3.2. Temporal-Spatial Variations in Multiple ESs*

3.2.1. Ecosystem Services Assessment

To reveal the temporal-spatial patterns and trends of ESs in the Danjiangkou reservoir area, three types of ESs, i.e., WY, CS, and HQ, were calculated using the InVEST model in 2000, 2008, and 2018 (Figure 4, Table 3). The average annual WY of Danjiangkou reservoir decreases from 2000 to 2018 (Figure 4), and the average annual WY was 192.23 mm, 78.59 mm, and 61.25 mm for the years 2000, 2008, and 2018, respectively (Table 3). The decline in WY was particularly obvious during 2000–2008, with an average decrease of 59%. WY shows spatially that the eastern region was the highest, followed by the central region,

while the northern and western regions were the lowest. The most significant decrease in WY concentrated in the eastern part of the Danjiangkou reservoir, but there were also small areas where the WY had increased, mainly in the regions with expanding construction land, as evapotranspiration of construction land is lower than other land use types.

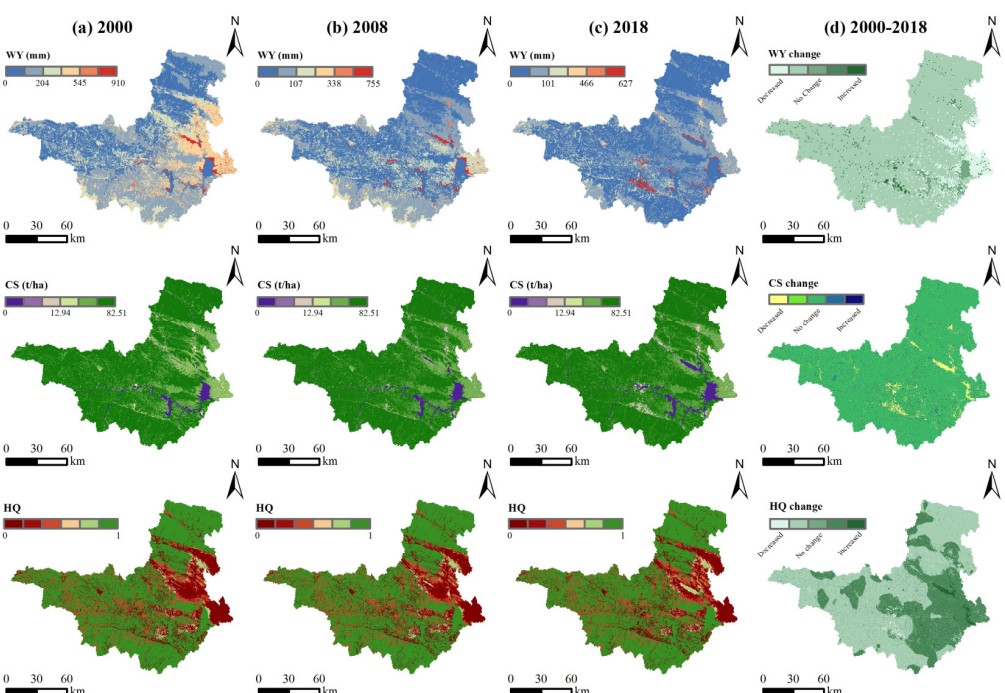

**Figure 4.** Spatial patterns of ecosystem services in the Danjiangkou reservoir area from 2000 to 2018.

**Table 3.** Average value of annual changes in ecosystem services in the Danjiangkou reservoir area from 2000 to 2018.

| Year | Water Yield (mm) | Carbon Storage (t/ha) | Habitat Quality |
|------|------------------|------------------------|------------------|
| 2000 | 192.23 | 72.64 | 0.83 |
| 2008 | 78.59 | 72.35 | 0.83 |
| 2018 | 61.25 | 70.93 | 0.83 |

Similarly, CS in the study area decreased during 2000–2018, with 72.64 t/ha, 72.35 t/ha, and 70.93 t/ha for 2000, 2008, and 2018, respectively (Figure 4). In the Danjiangkou reservoir area, CS mainly shows a spatial distribution of high values in the southern, western, and northern regions, and low values in the eastern region. Areas with increasing CS during 2000–2018 were scattered, while areas with decreasing CS were mostly concentrated in construction land and water area.

Overall, HQ index of the Danjiangkou reservoir area was high and remained stable during the study period (Table 3). Low HQ areas were concentrated in eastern regions, mainly occupied by construction and arable land, with more intensive human activity. Although the overall HQ in the Danjiangkou reservoir area did not change much, there were still some areas with decreasing or increasing HQ during the study period. HQ declined in areas where construction was expanding, whereas HQ improved in the northern and eastern parts.

### 3.2.2. Ecosystem Services Assessment

To further reveal the spatial clustering characteristics of each ES in the Danjiangkou reservoir area, we applied hot spot analysis. Figure 5 shows the distribution of cold spots and hot spots for each ES in the Danjiangkou reservoir area in 2000, 2008 and 2018. There

are differences in the distribution of cold spots and hot spots for each ES in the Danjiangkou reservoir area. In terms of the hot spots, the distribution of hot spots for WY was the most significant, and it was concentrated in the eastern part of the study area in 2000. The hot spots distributed in the eastern part of the study area decreased after 2000, but many new hot spots were added in the southern part of the study area in 2018. The hot spots of CS and HQ changed very similarly from 2000 to 2018. Hot spots for CS and HQ in 2000 were concentrated in the southern part of the study area, but the hot spots started to decrease after 2000 and shifted to cold spots in some areas by 2018. These areas have seen an increase in population and socio-economic activity and have been greatly affected by human activity. In terms of cold spots, the cold spot distribution of WY was not significant and it largely disappeared by 2018. The cold spots of CS and HQ were concentrated in the eastern part of the reservoir from 2000 to 2018.

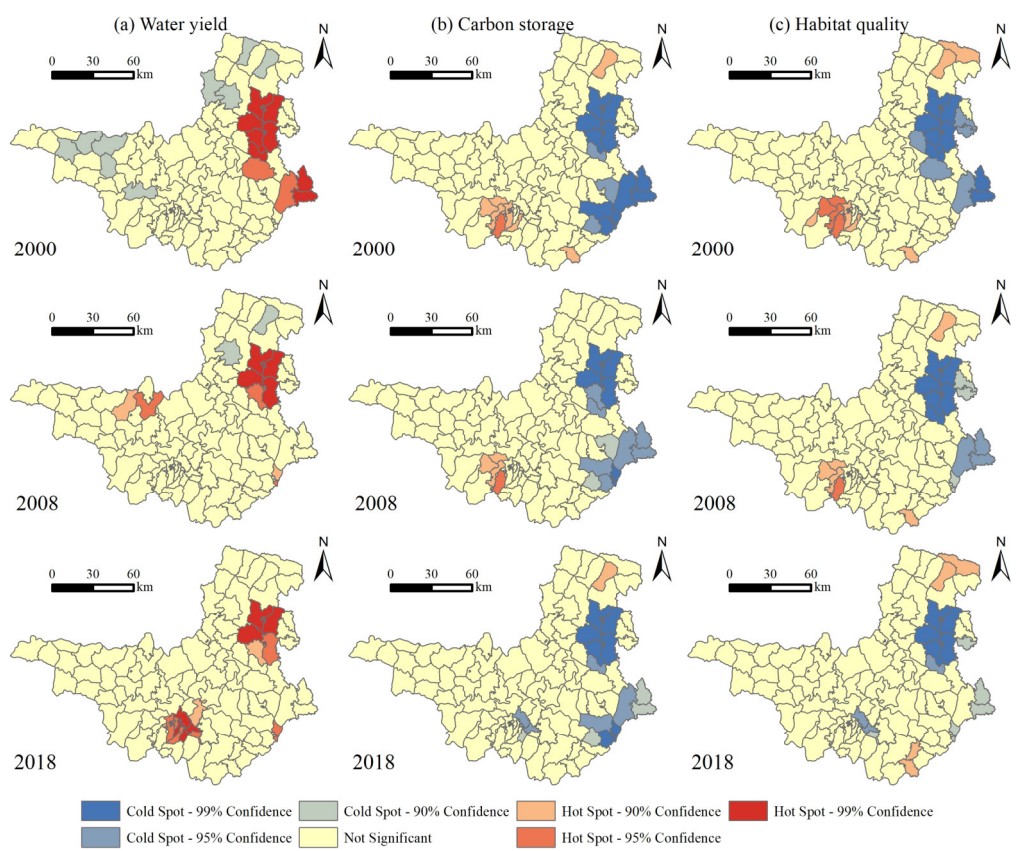

**Figure 5.** Hotspot distribution of ecosystem services in the Danjiangkou reservoir area from 2000 to 2018.

### *3.3. ESs Tradeoffs and Synergies*

To evaluate the correlation between the three ESs of WY, CS, and HQ, the Pearson correlation coefficient was applied at a township scale. As shown in Figure 6, there were significant tradeoffs and synergies between WY and CS, WY and HQ, CS and HQ.

Spatial correlation of WY and CS was mainly synergistic, with townships having positive correlations reaching 67.24% and being distributed throughout the reservoir area. There were also 29.31% of townships with negative tradeoff relationships along with a fragmented distribution. Regarding WY vs. HQ, 54.31% of the towns had a positive spatial correlation between WY and HQ, i.e., a synergistic relationship, located mainly in northern and central-western parts of the Danjiangkou reservoir area. The remaining 45.69% of villages and towns displayed a tradeoff relationship, and the vast majority were located in the eastern and central regions. Combined with the annual changes of ESs in Figure 4, it can be found that the areas with tradeoffs in the eastern part of the study area were mainly due to the change in land cover as a result of migration relocation. Infiltration of precipitation

and evapotranspiration increase, resulting in a decrease in WY. In contrast, the increase in reservoir storage and the relocation of migrants have improved the eco-environment and HQ of these areas. Therefore, the WY and HQ in these areas showed a tradeoff relationship. Furthermore, the interrelationship between CS and HQ was primarily synergistic, with 71.55% of towns having synergistic relationships, and 25% having trade-off relationships. The northern, southern and western regions of the reservoir area mostly showed synergistic relationships, and these regions were mainly dominated by woodlands and grasslands, which had good ecological environment and carbon sequestration capacity. As a result, CS and HQ had high values and a strong synergistic relationship. On the other hand, the tradeoff area was concentrated in the eastern part of the reservoir area, which is dominated by water areas, arable land and construction land.

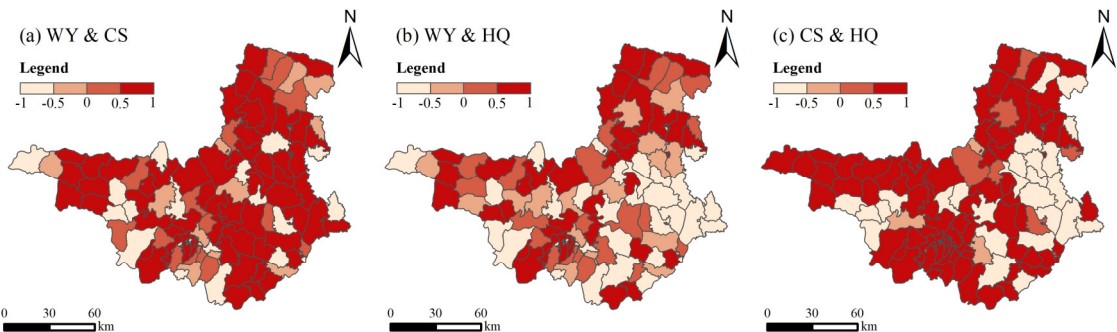

**Figure 6.** Tradeoffs and synergistic relationships of ecosystem services in the Danjiangkou reservoir area.

*3.4. Driving Factors of ESs*

According to the results of factor detection (Table 4), the strongest explanatory effects among the socio-economic factors were road network density and construction land share, both of which had explanatory power greater than 0.5 and was significant at the 0.01 level for all three types of ESs. In addition, the explanation degree of GDP on CS was greater than 0.4 and significant at the 0.01 level, which had a strong influence on CS. Among the natural factors, NDVI had the strongest explanatory effect on ESs. Meanwhile, other influencing factors had some degree of influence on ESs, e.g., WY was also influenced by the distance of towns to the nearest county, river network density, annual precipitation, and slope; HQ was influenced by GDP, river network density, annual precipitation, and slope; and CS was also influenced by annual precipitation.

**Table 4.** The degree to which the drivers explain the differences in ecosystem services in 2018.

| Detection Factors | | Water Yield | | Carbon Storage | | Habitat Quality | |
|---|---|---|---|---|---|---|---|
| | | *q* | *p* | *q* | *p* | *q* | *p* |
| Road network density | $X_1$ | 0.8285 *** | 0.000 | 0.6097 *** | 0.000 | 0.5920 *** | 0.000 |
| Construction land ratio | $X_2$ | 0.7636 *** | 0.000 | 0.6128 *** | 0.000 | 0.6235 *** | 0.000 |
| Distance to the county | $X_3$ | 0.3034 *** | 0.000 | 0.2463 | 0.1330 | 0.2337 | 0.1031 |
| GDP | $X_4$ | 0.2993 | 0.1780 | 0.4061 *** | 0.000 | 0.3309 *** | 0.000 |
| River network density | $X_5$ | 0.2197 *** | 0.000 | 0.1184 | 0.1122 | 0.1449 * | 0.0373 |
| Annual precipitation | $X_6$ | 0.1155 * | 0.0291 | 0.2643 * | 0.0238 | 0.2961 ** | 0.0044 |
| NDVI | $X_7$ | 0.3939 *** | 0.000 | 0.6036 *** | 0.000 | 0.4635 *** | 0.000 |
| Slope | $X_8$ | 0.1570 ** | 0.0034 | 0.3181 | 0.0553 | 0.2830 | 0.1079 |

Note: *** $p < 0.001$; ** $p < 0.01$; * $p < 0.05$.

## 4. Discussion

*4.1. Temporal-Spatial Evolution of LULC and ESs in the Context of the Central Line of SNWDP*

With the rapid urbanization and socio-economic growth, construction land expansion in the Danjiangkou reservoir area has led to the encroachment of a large amount of

woodland and arable land. From 2000 to 2018, arable land and woodland have gradually decreased, while construction land has increased by 1.31%. Most of the construction land expansion occurs in the southern region, particularly in Shiyan City, south of the reservoir. More importantly, LULC changes in the Danjiangkou reservoir are also impacted by national policies, including the SNWDP and the SLCP. Since 2005, when the top of the Danjiangkou Reservoir Dam was raised from 162 m to 176.6 m, the water area in the reservoir area has expanded significantly, flooding of a large amount of arable land. In addition, 348.92 km$^2$ of arable land was converted to woodland from 2000 to 2018. This is mainly due to the significant changes brought about by the long-term implementation of the SLCP. These changes show that, LULC can be shaped by relevant national projects and human activities.

The changes in LULC have triggered variations in multiple ESs provided by local ecosystems. ESs in the Danjiangkou reservoir area have decreased from 2000 to 2018, which is in line with the findings of Liu et al. [57], who found a decreased trend and a scattered pattern in eco-security of the Danjiangkou reservoir area. Among them, WY is the type of ES with the most significant decline. Combined with LULC, WY is decreasing for all land use types except for construction land, since evapotranspiration of construction land is significantly lower than ecological land use [60]. During urbanization and construction land expansion, the underlying surface changes and the impervious surface area increases, thus reducing evapotranspiration and precipitation infiltration. In addition, CS also showed decreases during the study period, with the most significant area of decrease being closely related to areas of expansion of water areas and construction land. This is attributed to the shrinkage of arable land and woodland due to the implementation of the SNWDP and the expansion of construction land, as woodland and arable are important sources of soil organic carbon [61]. While the HQ is generally stable, spatial variation is apparent. The eastern part of the reservoir area (Figure 3) has a higher concentration of arable land, making it more susceptible to human activity, resulting in lower HQ spatially. In addition, due to the SNWDP, there has been extensive ecological migration around the reservoir area in eastern areas, resulting in an increase in HQ from 2000 to 2018 as human activity decreases (Figure 4).

The findings of this study also demonstrate that both socio-economic and natural factors are responsible for the changes in ESs [62], and their effects differ across multiple types of ESs. Therefore, it is critical to investigate the impacts of different factors on ESs to develop regulatory measures for ESs in the Danjiangkou reservoir area. According to the GDM, we found that road network density and the proportion of construction land are the primary factors contributing to ESs reductions and spatial differentiation in the reservoir area. Among the natural factors, NDVI is the leading factor with the strongest influence on ESs changes, which is consistent with the findings of Zhang et al. [63]. NDVI is a commonly used vegetation index that monitors vegetation growth and reflects surface vegetation cover [64,65], and is positively correlated with CS and HQ. Consequently, we should focus on the self-regulating function of ecosystems in ecosystem management and control the unlimited expansion of human activities, thereby strengthening the protection of the ecological environment in the reservoir area and ensuring the ecological security of the core water source of SNWDP.

### 4.2. ESs Tradeoffs and Synergies

In the Danjiangkou reservoir area, WY, CS, and HQ exhibit both synergistic and tradeoff relationships due to climate change and LULC changes caused by SNWDP and human activities. In WY and CS, a high synergistic relationship results primarily from an increase in water area and storage. Water plays a critical role in linking and controlling many different ecosystems [66], and a relatively stable supply of water is essential to maintaining other ecosystems in a watershed or water source. In addition, the vegetation cover in the Danjiangkou reservoir area is high, which makes the vegetation transpiration of water consumption enhanced. As a result of construction land expansion, WY and HQ have both

tradeoffs and synergies. With the expansion of construction land in Danjiangkou reservoir, the ecological environment is severely damaged, resulting in a decrease in HQ, but the WY in these areas is enhanced by the barrier provided by the underlying surface cover. In addition, CS and HQ are primarily synergistic in northern, southern, and western regions since both are connected to woodlands and grasslands. Due to the complex tradeoffs and synergies among multiple ESs, flexible and efficient policy measures are essential to safeguard the water quality and subsequent plans for the core water source of the SNWDP.

*4.3. Policy Implications*

According to an in-depth analysis of LULC and the ESs of the Danjiangkou reservoir area, the following policy recommendations could be made to guide the subsequent planning and management of the Central Line of SNWDP and SLCP. Firstly, we found that the conversion area between arable land and woodland was equivalent during the study period, which indicates that the SLCP was accompanied by more destructive human activities, such as deforestation and clearing. In order to improve the overall ecological environment of the Danjiangkou reservoir area as well as soil and water conservation capacity, it is essential that the SLCP be further implemented and supervised, particularly along the reservoir area and in out-migrant areas.

Second, the raised Danjiangkou Dam has resulted in the expansion of water area, which had a great impact on other land use types, especially the arable land. Therefore, the red line for permanent basic farmland protection, the urban development boundary, and the ecological protection red line should be reasonably delineated in territorial planning to prevent unreasonable expansion of urbanization and to guarantee food supply and ecological security within the Danjiangkou reservoir area.

Finally, the core water sources of the SNWDP provide critical ESs to the water supply and receiving areas, not only in terms of the water resources and supporting water products they provide, but also other vital ESs such as water conservation, soil conservation, CS, and HQ. Due to the large spatial span of the SNWDP, spanning multiple watersheds as well as administrative regions, it is difficult to balance the costs and benefits of both ESs supply and demand, especially the core water sources, during the planning and decision-making process. Therefore, it is essential to establish a reasonable and scientific ecological compensation mechanism to alleviate the conflicts between the south and the north, and between the water supply area and the water receiving area during the implementation of the central line of the SNWDP, and to minimize the negative effects of water diversion.

*4.4. Limitations and Prospects*

This study examined the spatial and temporal variability of land use and ESs, identified tradeoffs and synergistic relationships among three types of ESs (i.e., WY, CS, and HQ), and revealed the driving factors of ESs change in the Danjiangkou reservoir area at a township scale. However, the relationships between multiple types of ESs, and its driving mechanisms vary across different scales. In the future, a comparative study of the ESs in the Danjiangkou reservoir area would be conducted at multiple scales. Furthermore, this study only quantitatively analyzed three typical ESs, so future studies can consider other ESs, such as soil conservation and water purification, to more comprehensively assess the regional ecosystem. Another limitation in the study is that we only classified the land use data into six main types without further subdivision of it, which may lead to a bias in the results. A more detailed classification of land use types should be adopted in future studies.

**5. Conclusions**

In this study, we explored the evolutions of land uses and ESs in the Danjiangkou reservoir area in the context of the implementation of national projects, including SNWDP and SLCP. Based on the InVEST model, hot spot analysis tool and Pearson correlation analysis, combined with spatial analysis tools, we investigated the temporal and spatial

dynamics, spatial clustering characteristics, and trade-offs and synergies of ESs in the Danjiangkou reservoir area. In addition, the GDM was used to detect the influence strength of natural and socio-economic factors on ESs and to identify the dominant factors influencing ESs in the Danjiangkou reservoir area. Results show that: during the period 2000–2018, the land use structure of the Danjiangkou reservoir area changed substantially, with the most notable of these changes being the significant expansion of the water area. As for ES variations, the HQ remains relatively stable, but spatial heterogeneity exists. Both WY and CS decreased during the study period, with the most pronounced decrease in WY. Variability in the LULC and climatic conditions are responsible for ESs variations. At the township scale, there is a significant spatial heterogeneity in the relationships of multiple ESs, with synergies mainly occurring in the northern, western and southeastern parts of the reservoir area, while tradeoffs are found in the eastern part of the reservoir area. In addition, both natural and socio-economic factors influence ESs in the Danjiangkou reservoir area, and the influence of socio-economic factors on ESs is dominant. These results indicated that the impact of socio-economic development should be fully considered when formulating ecological management policies to improve the overall level of ESs. However, the role of natural conditions should not be ignored in order to ensure the optimal development of the ecological environment in the Danjiangkou reservoir area.

**Author Contributions:** Conceptualization, L.L. and L.Z.; methodology, L.L.; software, L.L.; validation, L.L., Y.W. and L.Z.; formal analysis, L.L.; investigation, L.L. and L.Z.; resources, L.L.; data curation, L.L., L.Z., B.Z. and Y.B.; writing—original draft preparation, L.L. and C.L.; writing—review and editing, Y.W. and L.Z.; visualization, L.L., L.Z., B.Z. and Y.B.; supervision, Y.W.; project administration, Y.W.; funding acquisition, Y.W. All authors have read and agreed to the published version of the manuscript.

**Funding:** This research was supported by the National Natural Science Foundation of China (Grant No. 41901213), the Natural Science Foundation of Hubei Province (Grant No. 2020CFB856), and the Philosophy and Social Sciences Foundation of the Department of Education of Hubei Province (Grant No. 20G017).

**Data Availability Statement:** The LULC data were obtained from the Resources and Environmental Sciences and Data Center, Chinese Academy of Sciences (https://www.resdc.cn/ (accessed on 20 October 2022)). The ad-ministrative boundary data were obtained from the Resources and Environmental Sciences and Data Center, Chinese Academy of Sciences (https://www.resdc.cn/ (accessed on 22 October 2022)). China soil dataset were obtained from the National Tibetan Plateau Data Center (http://data.tpdc.ac.cn (accessed on 20 October 2022)). Potential evapotranspiration data were obtained from Global Aridity Index and Potential Evapotranspiration (ET0) Climate Database v2 (https://doi.org/10.6084/m9.figshare.7504448.v3 (accessed on 20 October 2022)).

**Acknowledgments:** The authors thank the National Natural Science Foundation of China (Grant No. 41901213), the Natural Science Foundation of Hubei Province (Grant No. 2020CFB856), and the Philosophy and Social Sciences Foundation of the Department of Education of Hubei Province (Grant No. 20G017) for support of this research.

**Conflicts of Interest:** The authors declare no conflict of interest.

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
