# Peer review of "Land Use and Ecosystem Services Evolution in Danjiangkou Reservoir Area, China: Implications for Sustainable Management of National Projects"

_land, doi:10.3390/land12040788_

Round 1

Reviewer 1 Report

The study explored land use changes and ecosystem service (ES) changes in the Danjiangkou reservoir area and the response of ecosystem services to natural and socio-economic factors in the context of the implementation of the Central Line of SNWDP. The research design is organized well and presented clearly. Here are several suggestions as follows. 1. It is better to present the novelty of the research and quick literature review in the abstract. 2. It is better to analyze the LULC change by year and the correlation analysis should consider the scale effect.  

Reviewer 2 Report

This an interesting and a solid quality paper that needs minor adjustments before acceptance. 

- Title is too long - should be more concise

- Abstract: lines 25-33, explain the details about results for this case study area. It would be better to have some more general comments about the applicability of your approach. I am referring to the method used: hotspot analysis tools and correlation analysis. You can provide a comment regarding their suitability for this type of research, as well as the input data one needs to collect before applying the methods - proposed approach.  

- Lines 110-121 belong to the Methods section 

- Figure 1 belongs to the Methods section 

- Line 151 forests, generally speaking, can be evergreen (coniferous), broadleaves (deciduous) or mixed. For mixed forests one of the first two categories can be dominant. In these terms, your description for vegetation should be modified

- Table 1 should include a footnote with abbreviations and full names of parameters. Even though, the abbreviations are provided in the above text, including them in a footnote will make a Table 1 self-contained, and this makes it easier for reader to keep up. 

- Table 1 precision of data (parameters) varies drastically, from 30 meters to 1 km. How would you comment this - is a 1 km resolution sufficient for these three parameters? Then, what happens with roads, rivers and borders - they are also included in the analysis, but the precision of measurements is not defined. 

- Figure 3: unused land does not exist in any of the figures (a, b and c) and therefore can be removed from the legend. 

- Table 2: modify title of the columns, first column is named "2000" and should be landcover. Again, unused land is equal to 0 all along and can be omitted. 

- Table 2: what does the "in-transfer" mean? 

- Lines 300-314, too much data. We can see these values in tables. Focus on comment about transforming the category of landcover, you provided an example arable land to woodland, you can add similar comments. 

- Table 3, what are the units

- Conclusion: Again, this an interesting approach that can be applied to numerous similar examples. Instead of providing detailed result for one case study area, it is better if you conclude about the methods used, and type of data needed for the analysis. 
